# Dynamic Acetabular Cup Orientation during Gait: A Study of Fast- and Slow-Walking Total Hip Replacement Patients

**DOI:** 10.3390/bioengineering11020151

**Published:** 2024-02-02

**Authors:** Ksenija Vasiljeva, David Lunn, Graham Chapman, Anthony Redmond, Lin Wang, Jonathan Thompson, Sophie Williams, Ruth Wilcox, Alison Jones

**Affiliations:** 1Leeds Institute of Medical and Biological Engineering, University of Leeds, Leeds LS2 9JT, UKlwang75@its.jnj.com (L.W.); s.d.williams@leeds.ac.uk (S.W.);; 2Leeds Institute of Rheumatic and Musculoskeletal Medicine, University of Leeds, Leeds LS2 9JT, UKgchapman2@uclan.ac.uk (G.C.);; 3National Institute for Health Research (NIHR) Leeds Biomedical Research Centre, Leeds LS7 4SA, UK; 4DePuy Synthes Joint Reconstruction, Leeds LS11 8DT, UK

**Keywords:** total hip replacement, acetabular cup, pelvic movement, walking speed

## Abstract

The dynamic orientation of total hip replacement acetabular cups during walking may vary substantially from their assumed position at surgical implantation and may vary between individuals. The scale of this effect is of interest for both pre-clinical device testing and for pre-operative surgical planning. This work aimed to evaluate (1) patient variation in dynamic cup orientation; (2) whether walking speed was a candidate proxy measure for the dynamic cup orientation; and (3) the relationships between dynamic cup orientation angles and planar pelvic angles. Pelvic movement data for patients with fast (20 patients) and slow (19 patients) self-selected walking speeds were used to calculate acetabular cup inclination and version angles through gait. For aim 1, the range and extremes of acetabular cup orientation angles were analysed for all patients. A large patient-to-patient variation was found in the ranges of both inclination angle (1° to 11°) and version angle (4° to 18°). The version angle was typically retroverted in comparison to the implantation position (greatest deviation 27°). This orientation is substantially different to the static, 0° version, simplifying assumptions in pre-clinical ‘edge loading’ testing. For aim 2, the cup orientation angles were compared between the fast- and slow-walking groups using statistical parametric mapping. The only significant differences observed were for cup version angle, during ~12% of the gait cycle before toe-off (*p* < 0.05). Therefore, self-selected walking speed, in isolation, is not a sufficient proxy measure for dynamic acetabular orientation. For aim 3, correlations were recorded between the acetabular cup orientation angles and the planar pelvic angles. The cup inclination angle during gait was strongly correlated (Spearman’s coefficient −1) with pelvic obliquity alone, indicating that simple planar assessment could be used to anticipate inclination angle range. The cup version angle was correlated with both pelvic rotation and tilt (Spearman’s coefficient 0.8–1), indicating that cup version cannot be predicted directly from any single pelvic movement. This complexity, along with the interaction between inclination angle and range of version angle, supports the use of computational tools to aid clinical understanding.

## 1. Introduction

The fifteen-year revision rate for primary total hip replacement (THR) surgeries in the UK is just 3–5% for patients over 75 years old. However, this number rises to 10–13% in the more active under 55s group [1]. The overall number of hip replacement surgeries continues to rise in all age groups, with around 20,000 primary surgeries performed on under 60s in 2019. In addition to the increasing financial burden of revision surgery, implant failures and subsequent surgeries carry the risk of patient morbidity and, in some cases, mortality. 

One of the causes for early THR failure is rim damage associated with sub-optimal acetabular cup orientation, which can be caused by either surgical malalignment [2], in vivo cup migration [3], patient anatomy [4], and kinematics [5], or combinations of these factors.

The acetabular cup component follows the pelvic movement during activities, meaning that pelvic kinematics affect the position of the cup relative to the femoral head component of the THR. The rim damage mechanism of ‘edge loading’ is believed to occur when there is an unloading and reloading of the joint, such as during gait. However, surgical planning approaches focus on static pelvic measures, while pre-clinical testing regimes simplify the contribution of the pelvis [6,7,8,9]. 

Clinically, the most common pre-operative planning and post-operative cup orientation measurements are performed using planar anterior–posterior (supine or standing) radiographs, which do not capture functional cup orientation in day-to-day patient life [10,11]. Additional static positions can be taken into account and used to assess hip joint orientation range in a more complete manner, such as the seated flexed position used in the Optimized Positioning System (OPS™) [12] and the seated and squat positions which have been analysed using a low-dose imaging system (EOS^®^) [13]. These methods allow for more comprehensive patient anatomical investigation, but are time consuming, require specialist equipment and expertise [14,15], and capture only static positions. Currently, it remains unknown whether alterations in three-dimensional, dynamic cup orientation during walking and other daily activities affect the risk of rim damage and whether dynamic cup orientation should be accounted for in pre-operative planning.

In industry, the most widely used pre-clinical in vitro tests [6,7,8] include a standardised gait simulation wherein the acetabular cup remains stationary throughout the gait cycle. While this is appropriate for wear testing under well-aligned and concentric bearing conditions, the contribution of pelvic movement to the relative alignment of the hip joint may contribute to the damage severity in edge loading scenarios [7]. In vivo hip joint kinematics and kinetics can be derived using data from a combination of marker-based motion capture and force platform systems. Using this system, it has been demonstrated that there are substantial differences between typical hip joint contact forces in vivo and those applied in pre-clinical wear testing [16].

The motion of the pelvis during walking has the potential to substantially affect the relative orientation of the hip joint components and therefore the severity of any edge loading mechanism. A better understanding of cup orientation during walking, and how that varies between individuals, may influence both standardised pre-clinical device testing regimes and the measures taken during surgical planning. Full marker-based gait assessment and musculoskeletal modelling is not available in all clinical centres and requires specialist expertise. Therefore, the correlation of trends in cup orientation with easily measured patient characteristics such as age, weight, or gait speed would provide an attractive proxy measure for both pre-clinical device testing and pre-operative surgical planning. It has been shown that hip joint contact forces can be differentiated based on patients’ walking speed [17]; lower-functioning patients with lower gait speeds displayed lower hip contact force compared to those who walked faster. The differences in force seen between the slow and fast walkers are generated by many factors, which include the movement of the pelvis. Therefore, walking speed could be hypothesised to be a reasonable candidate as a proxy for dynamic acetabular orientation. The individual angular pelvic movement components, tilt and obliquity, have been shown to vary with walking speed. Van Emmerik et al. showed that range of obliquity angle significantly increased (*p* < 0.001), and range of internal–external rotation decreased (*p* < 0.05), with increased walking speed [18]. Another study has indicated that pelvic tilt can be affected by walking speed, causing a decreased range of motion (*p* = 0.03) [19]. However, to the authors’ knowledge, the combined effect of all three pelvic movement components on the acetabular cup orientation has been given limited attention to date.

In this study, the three-dimensional pelvic movement and the resulting acetabular cup orientation were investigated across two groups with differing walking speeds. The aims of the study were to (1) examine the dynamic range in cup orientation during gait and the variations between individuals that could inform the cup orientation used in pre-clinical testing and to (2) examine whether there are relationships between dynamic cup orientation and either walking speed or pelvic movement components that could be used in pre-operative surgical planning.

## 2. Materials and Methods

### 2.1. Gait Data and Patient Selection

Raw kinematic gait data were acquired at Leeds Biomedical Research Centre as a part of the Life Long Joints (LLJ) cohort [20,21]. In the original cohort, 132 total hip replacement patients undertook 3–5 walking trials, at self-selected speed, along a 10-metre walkway. Ethical approval was obtained (IRAS 14/NE/1013), and all patients provided informed written consent to participate in the study [17]. For inclusion in that original cohort, an individual had to be between 1 and 5 years post-THR surgery; older than 18 years of age; have no lower limb joints replaced other than hip joint(s); and fully pain free and not suffering from any other orthopaedic or neurological problem which may compromise gait. Gait data were captured using a ten-camera Vicon system (Vicon Motion Systems Ltd., Oxford, Oxfordshire, UK), and the CAST marker set was used to track lower-limb kinematics in six degrees of freedom, with four non-orthogonal marker clusters positioned over the lateral thighs, lateral shanks, and sacrum as described comprehensively elsewhere [22].

From the LLJ cohort, data for 39 unilateral THR patients were selected for this study. Patients were sorted into the fast-walking group (n = 20) when their select selected walking speed was one standard deviation above the mean of the wider cohort [20] (resulting in speeds ≥ 1.26 ms^−1^). Likewise, patients were sorted into the slow-walking group (n = 19) when their speed was one standard deviation below the wider cohort mean (≤0.95 ms^−1^). The demographics of each group are given in Table 1.

Pelvic movement components were derived for each patient using Visual 3D (C-Motion, Inc., Germantown, MD, USA). The components were defined as pelvic tilt (rotation around a medial–lateral axis); obliquity (rotation around an anterior–posterior axis); and internal–external rotation (rotation around superior–inferior axis).

### 2.2. Computational Simulation

An algorithm was written in Python (3.7) to calculate the orientation of the acetabular component through the gait cycle for each patient. The codes are openly available in a software release [23].

The acetabular cup was assumed to be implanted at 45° inclination and 7° version for every patient [24]. In a neutral position, where pelvic tilt, obliquity, and rotation were all zero, the pelvic coordinate system was set to be identical to the laboratory coordinate system, and the anterior pelvic plane (APP) was parallel to the laboratory frontal plane. In this neutral pelvic orientation, the implanted cup orientation was the same in both systems.

The inputs for the algorithm were as follows: the cup orientation in neutral pelvis position, expressed as an inclination and a version angle, and the three pelvic orientation angles for each of the 101 points through the gait cycle. The outputs were the dynamic cup inclination angle (Figure 1a) and the dynamic cup version angle (Figure 1b), where the term ‘dynamic’ denotes measurement for every gait cycle point. The acetabular cup was represented by the outward facing normal vector of the acetabular cup rim plane, which was collinear with the vector between cup centre and cup pole. That outward facing normal vector was initially defined to comply with the inclination and version angles of the assumed implantation position (described above). For each point in the gait cycle, the vector was then rotated by applying the three pelvic angles in a Cardan sequence according to the pelvis segment joint angle calculation standard in Visual3D (C-Motion, Inc., Germantown, MD, USA) [25]. Specifically, the internal–external rotation was applied first, followed by obliquity and, finally, tilt.

### 2.3. Data Analysis

For each patient, the mean and range of the inclination and version angles were used. These were compared to the orientation at implantation (45° inclination, 7° version). The range and mean between two patient groups were also compared using a two-tailed *t*-test (α = 0.05).

A comparison of the two patient groups for the whole gait cycle was performed using the Statistical Parametric Map (SPM) [26] methodology. This method was chosen as it allows for the statistical analysis of the whole gait cycle as a continuous process rather than analysing discrete gait cycle points [27].

A sensitivity test was performed to evaluate the influence of the assumed cup orientation at implantation on the difference between patient groups. Four cup orientations were chosen for the test, representing the bounds of the “Lewinnek Safe Zone” [28]. The selected orientations were combinations of 30° or 50° inclination and 5° or 25° version. The dynamic cup orientation calculations were all repeated for each of these four assumed implantation orientations. The SPM comparison of the two patient groups was repeated for each of the four implantation positions.

Finally, Spearman’s rank correlation coefficients were calculated to establish any relationships between each pelvic angle (tilt, obliquity, and internal–external rotation) and each cup orientation measure (inclination and version). These calculations were performed separately for each patient group, resulting in a total of 12 comparisons. For a given correlation calculation (e.g., pelvic tilt against cup inclination), the two sets compared were composed of the mean values of that angular measure over the patient group at each data point in the gait cycle. The significance levels of the Spearman’s rank coefficients were checked using Student’s *t*-test distribution, and only results where the correlation coefficient was significantly different from zero (α = 0.05) are reported.

## 3. Results

The mean and standard deviation of three pelvic rotation angles through the gait cycle are presented in Figure 2 for the two patient groups to provide context for the rest of the results.

### 3.1. Patient-to-Patient Variation in Dynamic Cup Orientation

The results for the mean and range of the dynamic cup inclination and version during the gait cycle for each patient revealed that the role of pelvic movement is more patient-specific than group-specific (Figure 3). For all patients, the mean cup inclination was clustered around the implantation inclination angle, with 20 subjects above 45° and 19 subjects below 45°. For most of the patients (37 patients out of 39), the mean dynamic version angle was more retroverted than the implantation angle (of 7°). The range of dynamic orientation angles was different between patients and varied from 1° to 11° for inclination and from 4° to 18° for version. Changing the implantation position (in the sensitivity test) made a maximum difference of 1° to the range of inclination and 4° to the range of version.

In the most extreme cases, the cup reached a maximum inclination of 52° (representing an increase of 7° from the implantation angle) and a minimum version of −20° (representing a decrease of 27° the implantation position). Implantation positions with low inclination angles showed greater differences between implantation version angle and maximum version angle during gait. The greatest version angle deviations were 43° for the implantation case with 30° inclination and 5° version and 47° for the implantation case with 30° inclination and 25° version.

### 3.2. Comparison of Fast and Slow Walking Groups

A comparison of the two groups’ minimums, maximums, averages, and ranges showed no significant difference. From the SPM analysis, it was found that there was no significant difference in inclination angle between the two groups during the whole gait cycle (Figure 4a,c). However, a statistical difference was found for version angle before toe-off, between 38 and 50% of the gait cycle (Figure 4b,d).

The identical SPM analyses performed for the four alternative cup implantation positions in the sensitivity study produced similar results. The difference in inclination angle remained not significant, and for version angle, significance was recorded before toe-off but with slight deviation in the gait region (Table 2).

### 3.3. Correlation of Pelvic Angles and Dynamic Cup Angles

For both the fast and slow groups, the inclination angle was strongly correlated with obliquity (Table 3). However, for the version angle, the exact correlations observed differed between the fast- and slow-walking groups. For the fast group, both tilt and internal–external rotation were strongly correlated with version angle, but for the slow group, only internal–external rotation was strongly correlated with version angle.

## 4. Discussion

This study involved a virtual investigation of the variation in acetabular cup orientation during gait across fast-walking and slow-walking THR patients. The study’s results provide insight into the variation between individuals, of interest in defining representative conditions for pre-clinical device testing. In addition, the degree to which dynamic cup orientation can be predicted by either walking speed or pelvic movement components is of interest in pre-operative surgical planning.

### 4.1. Discussion of Findings

Pelvic movement during gait and its effect on the orientation of the acetabular cup was patient-specific. The patients with the lowest pelvic mobility had very small changes in cup orientation during gait (e.g., 1° change in inclination and 4° change in version), whereas those with the highest pelvic mobility could generate a range of 10° in inclination angle and 18° in version angle. This large *range* demonstrates substantial deviation from the implanted position during gait regardless of any contribution of postural pelvic tilt. The dynamic inclination angles were found to be clustered around the assumed implantation angle; in contrast, the dynamic version during gait was typically retroverted in comparison to the implantation position within the pelvis. The desired radiological cup orientation at implantation differs between surgeons and products. Harrison et al. [24] report recommended radiographic inclination angles between 30 and 50° and version angles between 0 and 23°. The dynamic change in orientation angles reported here increase that patient-to-patient variability. Extreme examples would be an implanted inclination angle of 50°, increasing to perhaps 55° during gait, or an implanted version angle of 0°, resulting in a cup which was retroverted by up to 18° around toe-off. These findings support those of Zheng et al. [28], who found many subjects whose cup orientation fell outside the Lewinnek Safe Zone during gait.

The finding of a high range of acetabular cup version angles during gait for some individuals has potential implications for the standardised testing of THR devices under edge loading conditions. Currently, the direction of separation and version angle are assumed to be consistent throughout the cyclic device testing. However, it is possible that an in vivo separation mechanism could interact with a large range of motion and high retroversion, resulting in more complex edge loading patterns, detrimental to the device lifetime. Further work is now needed to examine these scenarios in laboratory simulator tests.

Groups of patients differentiated by fast and slow self-selected walking speed were used to assess whether walking speed could be used as a proxy for cup orientation measures. Subjects with a large range of dynamic inclination and version angles during gait were registered in either slow- and fast-walking groups, and the difference in group range was not statistically significant. For the cup version angle, some significant difference was found before toe-off, and that difference persisted at different implantation positions. While the existence of a difference in version angle before toe-off may persist in a wider sample group, the high standard deviation within each group is also likely to be maintained. Therefore, these findings indicate that the speed alone cannot be used with any confidence to stratify patients or make assumptions about their pelvic movement and subsequent cup orientation.

When the relationships between dynamic cup orientations and pelvic movement components were examined, it was found that the dynamic inclination angle was strongly correlated with pelvic obliquity alone. This implies that individual patients who are likely to have large deviation in inclination angle could be identified through gait analysis in the absence of additional computational work.

However, the most substantial deviations from implantation orientation were seen in the retroversion of the cup during gait. The high deviations in version angle seen for some patients in this study suggest that pre-surgical clinical assessments of dynamic pelvic movement during activities may be appropriate. In the current study, the results derived from the sensitivity testing showed that when the implanted inclination angle was lower, the deviation and range of cup version angles were greater. This interaction of factors was also reported by Snijder et al. [29] for a large range of pelvic tilt angles in the sagittal plane. In the current study, the effect was replicated in the context of three-dimensional gait-based motion. For the fast-walking group, where the highest dynamic retroversion angles were found, it was shown that pelvic rotation contributed to cup version, alongside the pelvic tilt. This finding is consistent with studies examining static body positions [30]. The need to combine two pelvic rotations to predict the acetabular cup orientation motivates the use of mathematical tools such as those used by Snijder et al. [29] and in the current study.

A well-established approach to hip surgery planning takes into consideration various static measures [14] which have been associated with patient outcomes. For example, the commonly referenced Lewinnek Safe Zone defines a region of radiographic cup inclination and version which has been shown to reduce the chance of dislocation [31]. More recently, advances in software and clinical imaging have allowed for a more patient-specific approach to surgical planning, including the consideration of hip joint range of motion [12]. Patient-specific implantation targets are likely to be of most benefit in patients whose optimal cup position deviates substantially from the current recommendations; therefore, balancing the risk of different failure modes is challenging. This work has provided an indication of the dynamic deviation from implanted position for walking gait, which is of particular interest when considering patient-specific implantation outside the well-established zone.

### 4.2. Limitations

A limitation of this study is the fact that it involved the re-use of motion capture data from an existing study of THR patients where image data were not available to measure the individual cup orientation. This meant that an assumed implanted cup orientation, which was the same for all patients, had to be applied. This approach had the advantage of removing a confounding factor in the analysis of the pelvic movement effects. The implantation position sensitivity study was introduced to establish whether the main conclusions would change with different implantation positions. Importantly, this study showed little difference in the dynamic cup orientation range, indicating that the key findings hold across the clinically relevant range of cup implantation angles. Patients’ cup orientations would, however, be needed for any future specific-specific analysis to evaluate individual risk of edge loading, but that was not the purpose of this study.

In the process of marker data collection, a cluster approach was used to limit the impact of soft tissue artifacts [32], which have been shown to generate errors in hip joint angles (estimated to be 1° in flexion–extension, 1° adduction–abduction, and 5° in internal–external rotation) [33]. Those estimates indicate robustness in our pelvic tilt and obliquity measurements, where the scale of error is small in comparison to the range of angles, and some unavoidable uncertainty in pelvic rotation. This uncertainty should be taken into consideration in any future work attempting to establish the extremes of acetabular cup version for pre-clinical testing; however, the group averages are representative.

The subject-specific postural standing pelvic tilt was not considered in this work, as that information was not available. All dynamic pelvic angles were applied from an identical neutral pelvic orientation for all patients. Therefore, these data capture patient-specific effects but do not separate postural and dynamic factors.

The individuals included in the study were asked to walk, at a natural speed for them, across a gait laboratory. Subsequently selecting those with high and low walking speeds provided data on the pelvic motion patterns in those sub-groups. It is unclear whether the individuals would remain in same sub-groups if their average walking speed in daily life were used instead. Therefore, the results of the analysis conducted in this study cannot be directly extended to the prediction of subject-specific outcomes.

This study focused on walking gait and neglects other activities of daily living. Walking is a regular activity for most people, and it forms the basis of standardised THR device testing protocols, making it a sensible initial activity for investigating dynamic acetabular cup orientation. However, gait is unlikely to include the most extreme joint orientation scenarios of daily living.

## 5. Conclusions

This study indicated that the change in cup orientation during walking is highly variable from patient to patient, and in many cases, the version angle deviates substantially from the current assumptions made in standard pre-clinical ‘edge loading’ testing. This finding motivates further work to assess whether the inclusion of those motions would initiate further device damage. The results have shown that self-selected walking speed, in isolation, cannot be used as a proxy measure to anticipate acetabular cup orientation during gait. This work supports the use of pelvic obliquity alone to indicate the range of acetabular cup inclination angle during gait, which may be viable in some current clinical settings. However, the prediction of acetabular cup version angles has been shown to require information on both pelvic tilt and rotation, as well as the inclination angle. The use of computational tools to aid the understanding of the cup motion is therefore supported, and the outputs from those tools for larger groups could help to identify targets for future hip device testing standards and setting device indications.

## Figures and Tables

**Figure 1 bioengineering-11-00151-f001:**
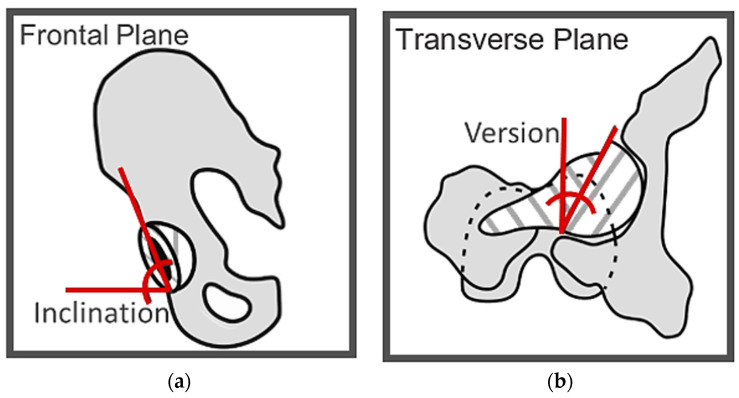
Illustration of the two acetabular orientation angles. (**a**) Pelvis with implanted acetabular cup in the frontal plane. Inclination was defined as the angle between the horizontal axis of the image and acetabular cup rim. (**b**) Pelvis and femur with implanted total hip replacement in the transverse plane. Version was defined as the angle between the vertical axis of the image and acetabular cup rim.

**Figure 2 bioengineering-11-00151-f002:**
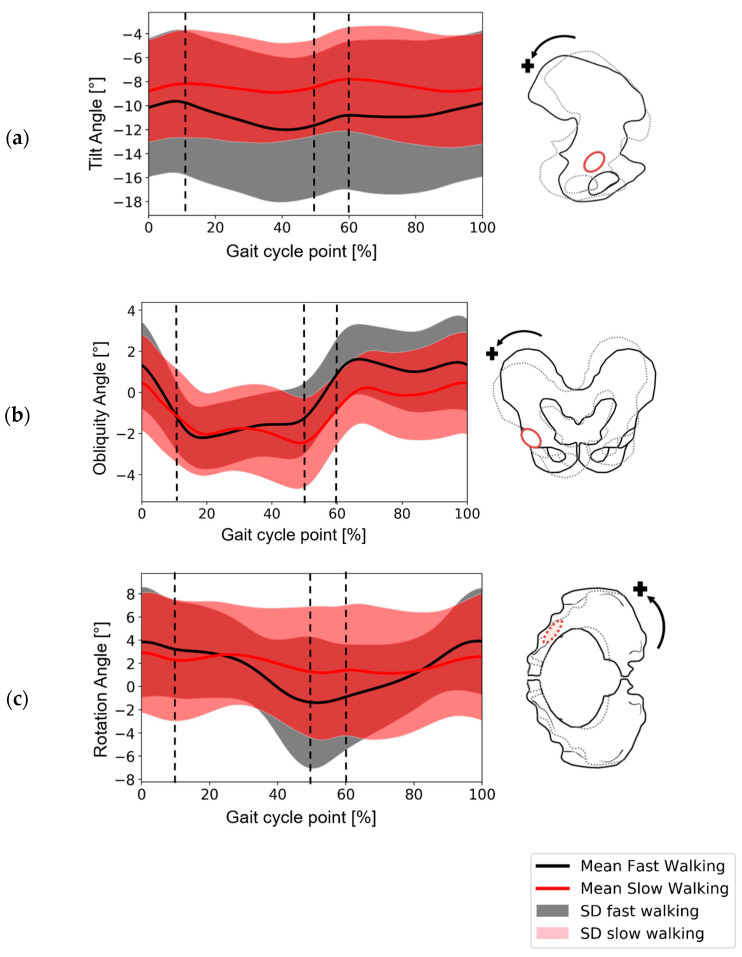
Derived pelvic movement data during gait for the fast-walking and slow-walking groups. The data have been normalised to the right hip according to right-hand rule. (**a**) Tilt angle around the medial–lateral axis (where the posterior tilt is positive)—sagittal plane. (**b**) Obliquity angle around the anterior–posterior axis (positive obliquity had downward direction)—coronal plane. (**c**) Rotation around the superior–inferior axis (where internal rotation is positive)—transverse plane. Vertical dotted lines indicate the approximate start and end of heel-strike (~0–10%) and toe-off (~50–60%).

**Figure 3 bioengineering-11-00151-f003:**
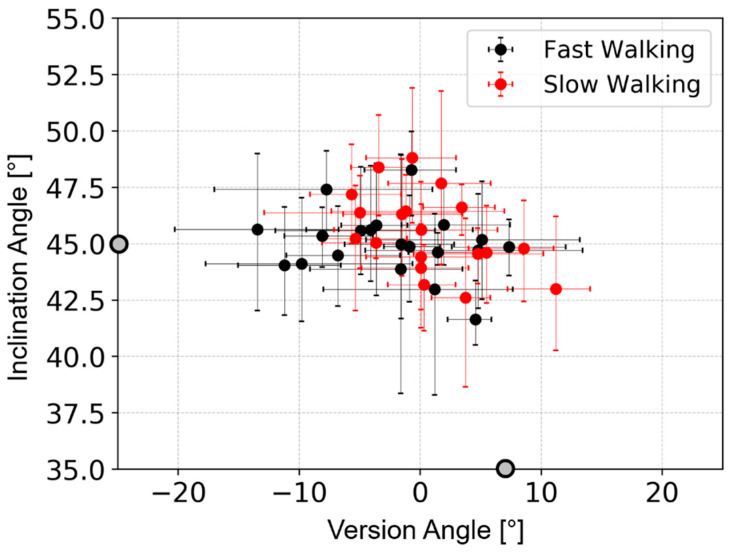
A plot of mean inclination angle against mean version angle for each patient. The mean is taken from all points in the gait cycle. The range of each angle is denoted by the bars for each data point. The fast-walking group is in black, and the slow-walking group is in red. The implantation angles of 45° inclination and 7° version are indicated by the grey dots on the axes.

**Figure 4 bioengineering-11-00151-f004:**
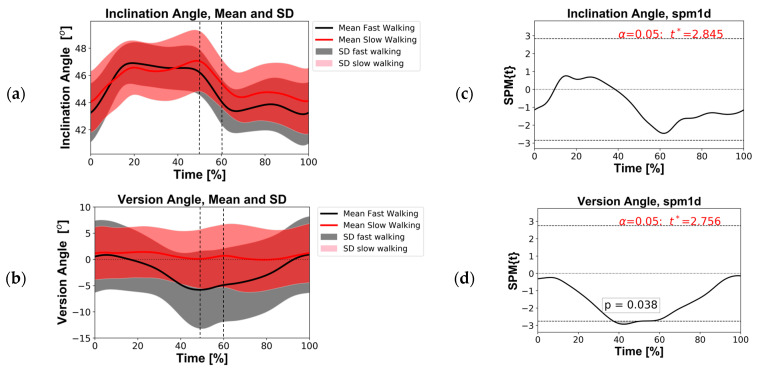
(**Left**) Angle profiles recorded over the gait cycle for the fast and slow groups. (**a**) Inclination angles; (**b**) version angles. The solid curves represent the group mean, and the transparent bands represent one standard deviation (SD). The vertical dotted lines indicate the approximate start and end of toe-off (~50–60%). (**Right**) Statistical parametric mapping (SPM) scalar trajectory *t*-test outputs. (**c**) Inclination angle; (**d**) for version angle. SPM{t} represent the *t*-test statistics continuum; when it exceeded threshold (t*, dashed lines top/bottom), significance (*p*-value) was recorded (α = 0.05).

**Table 1 bioengineering-11-00151-t001:** Patient demographics for the slow and fast groups.

Group	Sex	Age (Years) Mean (s.d.), Range	Body Mass Index (kg/m^2^) Mean (s.d.), Range
Fast	7 females, 13 males	68 (6.7), 57–81	28 (3.5), 22–35
Slow	10 females, 9 males	77 (6.2), 67–91	29 (4.3), 23–37

**Table 2 bioengineering-11-00151-t002:** Sensitivity SPM *t*-test (α = 0.05) results for five cup implantation scenarios.

Scenario: Inclination(^o^), Version (^o^)		45°, 7°	30°, 5°	30°, 25°	50°, 5°	50°, 25°
*p*-value [gait cycle points]	**Inclination**	NS *	NS *	NS *	NS *	NS *
	**Version**	0.038 [38–50]	0.034 [39–59]	0.042 [39–52]	0.043 [38–46]	0.047 [39–44]

* NS = not significant.

**Table 3 bioengineering-11-00151-t003:** Spearman’s rank correlation coefficient between dynamic cup orientations and pelvic movement components. The closer the coefficient is to +1 or −1, the stronger the correlation. Only results where the correlation coefficient was significantly different from zero (α = 0.05) are reported.

	Tilt		Obliquity		Rotation	
**Inclination**	Fast:	−0.4	Fast:	−1	Fast:	-
	Slow:	-	Slow:	−1	Slow:	-
**Version**	Fast:	+0.8	Fast:	-	Fast:	+1
	Slow:	-	Slow:	-	Slow:	+0.8

## Data Availability

The data associated with this paper are openly available from the University of Leeds Data Repository [34].

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
