# Peer review of "Dynamic Acetabular Cup Orientation during Gait: A Study of Fast- and Slow-Walking Total Hip Replacement Patients"

_bioengineering, 2024, doi:10.3390/bioengineering11020151_

Round 1

Reviewer 1 Report

Comments and Suggestions for Authors

This manuscript entitled “Dynamic acetabular cup orientation during gait: a study of fast and slow walking total hip replacement patients” investigates the calculation of acetabular cup inclination and version angles during walking gait. The reviewer believes that the authors are simply describing the whole part of the method. By "…by… or using…to…", such as Lines 15-17: the relationships between dynamic cup orientation angles and planar pelvic angles. Pelvic movement data for patients with fast (n=20) and slow (n=19) self-selected walking speeds, was used to calculate acetabular cup inclination and version angles through gait; Line 119: Gait data was captured using a ten-camera Vicon system (Oxford Metrics, UK) and the CAST marker set was used to track lower limb kinematics. Lines 127-128: The cup was rotated by applying three pelvic movement components in Cardan sequence according to the pelvis segment joint angle calculation standard in Visual 3D. The author neither gives a detailed description nor cites the relevant literature. The reviewer is very worried about this because it directly affects the quality of the article and readers' understanding of the article. At the same time, as the author mentioned in the limitation part, there are big problems with the current research method, which is also the concern of the reviewer. Specific comments are shown below:

Abstract

1. Lines 12-13: The background section provided by the author seems insufficient. Please revise.

2. Lines 23-24: “The cup version angle was correlated with pelvic rotation for both fast and slow groups, and with both pelvic rotation and tilt for the fast-walking group.”

Please make the comparison between the brisk walking group and the slow walking group clearer. Lines 24-26: Clearly state the hypothesis that your study is designed to test. For example, if you hypothesize that some dynamic version angles deviated from the implanted position enough to prompt a review of the simplifying assumptions in pre-clinical ‘edge loading’ testing and perhaps to affect the optimal surgical choices for those patients. Then state this clearly.

Introduction

4. Lines 33-34: The fifteen-year revision rate for primary total hip replacement (THR) surgeries in the UK is just 3-5% for patients over 75 years old. The experimental method did not explain whether the patient's acetabular cup was revised.

5. Lines 85-86: “The individual angular pelvic movement components, tilt and obliquity, have been shown to vary with walking speed.” This needs to be described in detail.

6. Based on prior research, it appears that the author has not formulated reasonable hypotheses for the experiments. Please provide supplementary information in this regard.

Materials and Methods

7. Line 99: Kindly include the criteria for participant inclusion and exclusion in the study.

8. Line 100: The methodology section lacks a clear elucidation on how the self-selected speed was defined by the author. Please request clarification on this aspect.

9. Lines 102-103: The reviewer suggested that the author clearly describe the specific location of the reflective markers, so that the following readers can directly refer.

10. Lines 106-107: How to determine “fast walking” and “slow walking”?

11. Line 144: Please refine the form, there is no standard deviation.

12. Lines 148-149: How do authors “takes this into account”?

Results

13. Lines 214-215: the authors wrote “However, for version, the results … groups”. I don’t know what the author wants to express.

Discussion

14. Line 324-325: As the author mentioned in the limitation part, there are big problems in the current research method, which is also the concern of the reviewer.

15. As the author mentioned in the abstract part, “Some dynamic version angles deviated from the implanted position enough to prompt a review of the simplifying assumptions in pre-clinical ‘edge loading’ testing and perhaps to affect the optimal surgical choices for those patients” (Lines 24-26). However, the reviewer did not see relevant discussions in the discussion section that reached this conclusion.

Conclusion

16. Line342-344: The authors need to emphasize the importance of this study, especially in terms of focusing on the change in cup orientation during walking, dynamic pelvic orientation, and prediction of the range of acetabular cup version angle during gait, which helps to highlight the practical value of the research.

17. The reviewer considers the author to be making recommendations for future research to delve into specific aspects of total hip replacement (THR), or to conduct related research in the broader field of sports medicine.

Comments on the Quality of English Language

no

Reviewer 2 Report

Comments and Suggestions for Authors

The manuscript "Dynamic acetabular cup orientation during gait: a study of fast and slow walking total hip replacement patients" investigated three-dimensional pelvic movement, and the resulting acetabular cup orientation, across two groups with differing walking speeds.

1. Please include some of the major results in the abstract, and the prominent conclusion also should be included.

2. The materials and methods section should be elaborated.  

3.The load conditions during the gait cycle should be clearly described and included in the computational section. 

4. Suggest to graphically represent the gait cycle used in the study.

5. Suggest to give a detailed mathematical and statistical analysis of the data analysis. Clearly describe various parameters taken into consideration for the data analysis.

6. Suggest to have results and discussion side-by-side rather than having them in two different sections.

7. The conclusions should be re-written in a concise and crisp manner. List the major conclusions in 3-4 bullet points.

8. The general standard of English is good. however there are sentence construct errors occasionally.

Comments on the Quality of English Language

 The general standard of English is good. However there are sentence construct errors occasionally. Check for typo errors.

Reviewer 3 Report

Comments and Suggestions for Authors

It is always a pleasure to read a well-organized and written manuscript. I have a few comments to help with clarification and organization (see pdf).

Round 2

Reviewer 1 Report

Comments and Suggestions for Authors

With some necessary modifications, the quality of Manuscript has improved significantly. However, the reviewer has the following major concerns about the limitations of this study:

1. Lack of individualized cup orientation information: Due to the use of motion capture data from a previous study of THR patients and the lack of image data to measure individual cup orientation, the same hypothetical implanted cup orientation was used in the study for all patients. This may lead to challenges in the applicability of study results at the level of specific individuals.

2. Impact of soft tissue artifacts: A cluster approach was used in the marking data collection to limit the effect of soft tissue artifacts. However, the paper mentions that soft tissue artifacts may produce errors on the hip angle that have not been discussed and evaluated in detail.

3. Neglect of specific individual differences: The paper notes that the lack of information on individual cuvette orientation precludes specific analyses of specific individual effects. The reviewers felt that this may have limited the individualized applicability of the study's findings.

Comments on the Quality of English Language

no

Round 3

Reviewer 1 Report

Comments and Suggestions for Authors

no

Comments on the Quality of English Language

no